# Dissection of the MKK3 Functions in Human Cancer: A Double-Edged Sword?

**DOI:** 10.3390/cancers14030483

**Published:** 2022-01-18

**Authors:** Valentina Piastra, Angelina Pranteda, Gianluca Bossi

**Affiliations:** Oncogenomic and Epigenetic Unit, Department of Diagnostic Research and Technological Innovation, IRCC—Regina Elena National Cancer Institute, Via Elio Chianesi 53, 00144 Rome, Italy; valentina.piastra@ifo.gov.it (V.P.); prantedaangelina@gmail.com (A.P.)

**Keywords:** MKK3, p38 MAPK, oncogene, tumor suppressor, cancer, therapeutic treatments, target therapy

## Abstract

**Simple Summary:**

Cancer target therapy urges the identification of novel molecular players to design novel strategies to improve the treatment outcomes by reducing the doses of conventional chemotherapy and preserving the effectiveness of the therapy. The Mitogen-activated protein kinase kinase 3 (MKK3) is an evolutionarily conserved protein kinase involved in the regulation of a plethora of cellular processes, however being not frequently mutated in human cancer the consequences of its dysregulation in gene expression or protein activity in cancer is controversial. With the aim to define a clear overview of MKK3 contribution in cancer a systematic search of recent literature focusing on MKK3 dissected functions was performed. The results revealed that the oncogenic and tumor suppressive functions of MKK3 are closely dependent on the tumor type, thus suggesting MKK3 as new putative molecular target to be enrolled for the design of more efficient anticancer therapeutic approaches for specific type of tumors contributing to add new avenues to be pursued to implement the effectiveness of precision medicine.

**Abstract:**

The role played by MKK3 in human cancer is controversial. MKK3 is an evolutionarily conserved protein kinase that activates in response to a variety of stimuli. Phosphorylates, specifically the p38MAPK family proteins, contribute to the regulation of a plethora of cellular processes such as proliferation, differentiation, apoptosis, invasion, and cell migration. Genes in carcinogenesis are classified as oncogenes and tumor suppressors; however, a clear distinction is not always easily made as it depends on the cell context and tissue specificity. The aim of this study is the examination of the potential contribution of MKK3 in cancer through a systematic analysis of the recent literature. The overall results reveal a complex scenario of MKK3′s involvement in cancer. The oncogenic functions of MKK3 were univocally documented in several solid tumors, such as colorectal, prostate cancer, and melanoma, while its tumor-suppressing functions were described in glioblastoma and gastric cancer. Furthermore, a dual role of MKK3 as an oncogene as well as tumor a suppressor has been described in breast, cervical, ovarian, liver, esophageal, and lung cancer. However, overall, more evidence points to its role as an oncogene in these diseases. This review indicates that the oncogenic and tumor-suppressing roles of MKK3 are strictly dependent on the tumor type and further suggests that MKK3 could represent an efficient putative molecular target that requires contextualization within a specific tumor type in order to adequately evaluate its potential effectiveness in designing novel anticancer therapies.

## 1. Introduction

The Mitogen-activated protein kinase 3 (MKK3, MAP2K3) is a member of the dual specificity protein kinase group (MKK) belonging to the mitogen-activated protein kinase pathway (MAPK). The MKKK proteins (MKK 1–4) activate MKK3 by phosphorylation at serine and threonine residues (Ser189; Thr193), in response to different cellular stress, as exposure to UV damage, DNA damaging agents, oxidative stress, as well as growth factors and cytokines [1,2]. Activated MKK3 and MKK6 are specific and direct upstream activators of the p38MAPK family proteins (α, β, γ, δ). The activated MAPK signaling mediates various cellular processes to respond and adapt accordingly. Indeed, other MAPKs such as p38MAPK proteins orchestrate cellular response by modulating a wide variety of targets, such as protein kinases, phosphatases, cell-cycle regulators, and transcription factors, including p53 [1,2,3]. Upon the interaction with multiple transcription factors in a tissue-specific manner, they also regulate proliferation, differentiation, apoptosis, invasion, cellular migration, and other cellular processes [1]. The exposure to chemicals, UV radiation, growth factors, and the abnormal activities of many signaling pathways regulating the normal cellular homeostasis are involved in the complex series of events leading to carcinogenesis, including proto-oncogenes and tumor suppressors mutations. Tumorigenesis is the consequence of consecutive somatic mutation accumulation that forces the selection of cells with acquired capability to grow out of control, promoting angiogenesis invasion spreading to the other body parts [4]. This evolutionary process involves gain-of-function and loss-of-function mutations in oncogenes and tumor-suppressing genes, respectively. Although genes, in carcinogenesis, are classified as oncogenes and tumor suppressors, a clear distinction is not always possible as it depends on the cell context and tissue specificity. Indeed, some genes exhibit both tumor suppressor and oncogenic functions in different tumors, suggesting their dual role in tumorigenesis under different cellular contexts. The majority of these genes encode multiple isoforms, which are further post-translationally modified and form a variety of protein complexes, generating a context-dependent cellular network [5].

MKK3 was first described in 1996 [6], and since then, almost 700 studies have been published focusing mainly on MKK3′s well-known function as the main activator of p38 MAPK signaling. However, recent literature explored novel MKK3 functions, such as the ability to interact with several partners, identifying MKK3 as a hub protein [7]. MKK3 is not frequently mutated in human cancer, and the consequences of its dysregulation in gene expression or protein activity in cancer are still controversial. Ongoing studies suggest MKK3 as an oncogenic player identified among the upregulated target genes by p53 gain-of-function mutants [8]. Its specific knockdown significantly affects proliferation and survival in cancer lines of different tumor types (breast, colon, melanoma), both in vitro and in vivo, inducing autophagy and cell death [1,8,9]. Accordingly, to provide a comprehensive overview of the MKK3 functions in human cancers, we systematically analyzed the recently published literature (last 20 years), highlighting its oncogenic and oncosuppressive reported functions in the different tumor contexts. Pubmed search for “MKK3 & cancer” (https://pubmed.ncbi.nlm.nih.gov/?term=%22mkk3+%26+cancer%22&sort=pubdate, (accessed on 25 November 2021) or “MAP2K3 & cancer” (https://pubmed.ncbi.nlm.nih.gov/?term=%22MAP2K3+%26+cancer%22, (accessed on 25 November 2021) identified 158 and 102 manuscripts from 1996 to 2021, respectively. Fifty-five of all analyzed manuscripts reported specific MKK3 functions, which interestingly revealed for the first time a more complex picture of MKK3′s role in tumor malignancy. Indeed, the oncogenic and tumor-suppressing functions of MKK3 were clearly dependent on the tumor type, suggesting that the evaluation of MKK3 as a potential molecular target in cancer therapy requires it to be contextualized with the organ affected by cancer disease.

## 2. Oncogenic MKK3 Functions

### 2.1. Breast Cancer

In breast cancer (Figure 1A), Huth et al., correlated the MEK2 and MKK3/6-p38MAPK axes activation with cyclin D1 deregulation. MEK2 silencing hampers the MKK3 activation downregulating cyclin D1 and inducing apoptosis in MDA-MB-231 cells [2]. Park et al., and Shin et al., studies described the crosstalk between RAS family members and MKK3. Indeed, the metastatic phenotype in nonmalignant breast epithelial cells (MCF-10A) induced by RAS ectopic expression was dependent on the expression of metalloproteinase-2 (MMP-2), which was sustained by activated MKK3 [3,10]. Xu et al., demonstrated that high receptor tyrosine kinase-like orphan receptor 2 (ROR2) triggers cell proliferation, migration, invasion, and epithelium–mesenchymal transition (EMT) via MKK3 activation in MDA-MB-231 and MCF-7 cells [11]. Indeed, the ROR2-induced MKK3 activation triggers the expression of genes associated with tumor invasion, such as transforming growth factor-β (TGFβ), MMP-2, and MMP-9 [11]. A wide variety of signaling pathways involve MKK3 as signal transduction hub; Jung et al., reported that the translationally controlled tumor protein (TCTP) sustained the activation of MKK3/6-p38MAPK signal pathway by interacting with the Na,K-ATPase α, inducing Src activation and Src release from Na,K-ATPase α subunit, promoting cell migration and invasion in MCF10A cells [12]. Interestingly, the TCTP depletion induces tumor reversion in lines expressing high levels of endogenous TCTP (MCF-7, T47D) [13]. Our studies demonstrated that MKK3 is required to sustain proliferation and survival in different lines (MDA-MB-468, MDA-MB231, SKBR3, MCF7) [8] in vitro. Interestingly, MKK3 appeared dispensable in the normal counterpart since its depletion was well-tolerated in nonmalignant breast epithelial cells (MCF10A) [9]. Our study demonstrated that TET-inducible MKK3 depletion induces autophagy and cell death and enhances therapeutic effectiveness when combined with chemotherapeutic agents (Adriamycin) in MDA-MB468 cells [9]. In accordance whit the MKK3′s oncogenic role in breast cancer, using a high-throughput screening approach in MCF7 cells to identify cancer-associated protein–protein interactions (PPI), Ivanov and co-authors identified MKK3 as a novel hub protein due to its interaction with several partners, including the oncogenic transcription factor c-Myc. Their study allowed the identification of MKK3 as a novel MYC regulator since its interaction inhibited MYC degradation by enhancing MYC-driven cell proliferation [7]. A recent study by Yang et al., conducted a systematic analysis of gene expression (mRNA) and clinical outcomes in TCGA breast cancer patients (n.1055 samples). It concluded that the MKK3 overexpression dramatically correlated with the worsened clinical outcomes in triple-negative African American breast cancer (TNBC) patients [14]. Their findings suggest that MKK3 drives epithelial–mesenchymal transition through sustained MYC activities.

### 2.2. Cervical and Ovarian Cancer

Peng et al. [15] and Yang et al. [16] reported the downregulation of microRNA-214 (miR-214) in cervical cancer (Figure 1B) as one of the mechanisms triggering excessive proliferation and tumor invasion through the MKK3 deregulation. The studies identified complementary sequences linking miR-214 to the MKK3 mRNA 3′UTR. The reduced miR-214 levels, detected in cervical carcinoma tissues, lead to a failure in MKK3 mRNA translation inhibition. In contrast, miR-214 overexpression reduces the MKK3 levels inhibiting the MKK3-p38MAPK signaling pathway and leading to cancer cell death [15,16]. Kumar et al., correlated the MKK3 activation with the high levels of Osteopontin (OPN) in cervical cancer tissues. Indeed, OPN drives cellular invasion through the CD44-dependent MKK3 phosphorylation, promoting the NF-kB-dependent transcription of the extracellular protease furin [17]. MKK3 inhibition reduces furin expression and cell motility [17]. Kim et al., found MKK3 to sustain cell motility and angiogenesis through essential molecular key players (MMP-2, MMP-9, Cdk4, Cdk2, and integrin β1) in SKOV-3 cells [18]. Kang et al., demonstrated that ectopic MKK3 drives paclitaxel resistance by inducing p-glycoprotein expression in Heya8 and Skov3ip1 cells, suggesting the involvement of MKK3-p38MAPK pathway activation in ovarian cancer drug resistance [19].

### 2.3. Liver Cancer

In liver cancer, Wang et al., correlated high levels of SIRTUNI1 (SIRT1) with increased p38MAPK activation through the induced MKK3 upregulation via YAP in Bel-7402, SMMC-7721 hepatocellular cancer lines (Figure 2A) [20]. The results show that SIRT1 silencing abrogates MKK3 expression and activation by inhibiting the transcription factor YAP, leading to cell death [20]. Luo et al., linked the overexpression of Sperm-associated antigen 9 (SPAG9) with excessive activation of the MKK3-p38MAPK signaling cascade in HEPG2 cells [21]. They found that SPAG9 depletion inhibits cell cycle progression, migration, and invasion by activating the MKK3-p38MAPK molecular axis [21].

### 2.4. Melanoma

In melanoma, Zhou et al. [22] correlated the low miR-21 levels with MKK3 upregulation in clinical samples from melanoma patients. Mir-21 targets the MKK3 mRNA 3′UTR; ectopic mimic miR-21 reduces proportionally and significantly MKK3 protein levels inducing cell cycle inhibition and death. In contrast, miR-21 inhibition increases MKK3, thus reducing apoptotic cell death and sustaining migration and invasion [22]. These findings suggested that the dysregulation of MKK3 activities acts as a mechanism to promote melanoma progression towards metastatic invasion. The oncogenic roles of MKK3 were also reported by Yi et al., who showed the L1 cell adhesion molecule (L1CAM)-dependent dysregulation of MKK3-p38MAPK signaling pathway activation in L1CAM highly expressing B16F10 melanoma cells. L1CAM siRNA silencing suppresses MKK3-p38MAPK pathway activation, thereby hampering B16F10 cells migration and invasion [23].

### 2.5. Prostate Cancer

In prostate cancer, Misra and Pizzo reported that the activated cell surface protein BiP (GRP78) supports proliferation through the MKK3-p38MAPK/α2-macroglobulin inhibitor protein (α_2_M*) crosstalk in 1-LN cells. The inhibition of GRP78 autophosphorylation hampers MKK3-p38MAPK signaling activation, triggering apoptosis [24]. Son J.K. et al., reported the MKK3 activation as a novel mechanism of resistance to tumor necrosis factor-related apoptosis-inducing ligand (TRAIL). TRAIL treatments upregulate the anti-apoptotic protein MCL-1 via the TAK1/MKK3/6-p38MAPK pathway activation, hindering cell death. Inhibition of p38MAPK downregulates MCL-1 restoring sensitivity to TRAIL treatments and inducing apoptosis in DU-145 cells [25]. Fan S. et al., further described novel MKK3 roles in driving resistance to Adriamycin in DU-145 cells, involving the over-expression of scatter factor (SF) and its c-Met receptor via src activation. Src involves the Rac1/MKK3/6-p38MAPK signaling pathway activation culminating in the overexpression of NF-κB transcription factor promoting cell survival and resistance to Adriamycin [26].

### 2.6. Esophageal Cancer

In esophageal cancer, Xie et al. [27] reported that Gossypetin inhibits the MKK3/6-p38 MAPK signaling pathway, hampering cell proliferation and motility and promoting apoptosis in different esophageal tumor lines (KYSE30, KYSE410, KYSE450, KYSE510), and patient-derived tumor tissue [27]. Nevertheless, more focused studies are required to clearly define the MKK3 oncogenic roles in this tumor type since most of the adopted experimental procedures investigated both MKK3 and MKK6 proteins.

### 2.7. Lung Cancer

In non-small-cell lung cancer (NSCLC) (Figure 2B), Yeung et al., reported the activation of the YAP-MKK3/6-p38MAPK–STAT3 signaling pathway in the gefitinib-induced genomic instability involving transient tetraploidization [28] and gefitinib-resistance in NSCLC cells (HCC827GR, H1975). The inhibition of p38MAPK (GW856553X, SB203580) restores sensitivity to gefitinib, eliminating tetraploidization and reducing STAT3 phosphorylation and p21, cyclin D1 expression, thus hampering the anchorage-independent cell proliferation in gefitinib-resistant cells [28]. Overall, no specific inhibitors or siRNA to each signaling molecule have been used to validate this study’s novel identified signaling pathway. Ko et al., described another molecular mechanism of gefitinib-resistance involving the MKK3/6-p38MAPK signaling pathway in NSCLC cells. They found that the inhibition of p38MAPK counteracts gefitinib-dependent expression of human Muts homolog 2 (MSH2) by increasing its cytotoxicity in NSCLC cells (H520, H1703) [29]. Tung et al. [30] demonstrated that pemetrexed increases MSH2 mRNA and protein levels in a manner dependent on MKK3/6-p38MAPK signal activation in NSCLC cells (H520, H1703). The MKK3/6-p38MAPK signaling activation prevents the ubiquitin-26S proteasome-mediated proteolysis of MSH2, reducing pemetrexed-induced cytotoxic effects. The inhibition of either MSH2 or p38MAPK markedly increases the NSCLC cell’s sensitivity to pemetrexed [30]. Tsai et al. [31] and Tseng et al. [32] studies demonstrated that the activation of MKK3/6-p38MAPK signaling pathway by Etoposide (VP-16) [30] or paclitaxel [31] is involved in the increased expression of Excision repair cross-complementary 1 (ERCC1) in A549, H1975 [31] and H1650, H1703 [32] NSCLC lines respectively, driving survival and resistance to DNA damage agents. The inhibition of p38MAPK (SB202190, siRNA) abrogates the drug-induced ERCC1 expression increasing NSCLC cell killing [31,32]. Galan-Moya et al. [33] proposed the imbalance between MKK3/6 as a novel biomarker to predict cisplatin therapeutic efficacy in NSCLC patients [33]. Indeed, by analyzing a panel of seven NSCLC derived lines (H23, Hop62, H157, H226, H460, H661, and H1299), they demonstrated that lines with high MKK3 protein levels are less responsive to cisplatin through a constitutive hyperactivation of the p38MAPK pathway, which in turn inhibits the MKK6 mRNA levels, suggesting in tumor cell-context, the existence of a regulatory mechanism only described in nontransformed MEF cells [34].

### 2.8. Colorectal Cancer

Colorectal cancer (CRC) studies performed in our laboratory support the MKK3 oncogenic role findings. As in breast cancer, we reported mutant p53 R273H to increase the MKK3 mRNA in the HT29 CRC line [8]. A survey conducted in a cohort of n. 189 CRC patients identified MKK3 as a poor prognostic marker since high levels of MKK3 significantly correlated with short overall survival (OS) in late-stage CRC disease [1]. Notably, depletion of the endogenous MKK3 strongly impaired HT29 cell proliferation and survival, inducing a significant G2/M phase accumulation [8]. Interestingly, MKK3 depletion induces autophagy and cells death in both mutant (HT29) and wtp53 (HCT116) bearing CRC lines, suggesting a more generalized effect. MKK3 depletion affects HT29 xenograft tumor growth in nude mice and potentiates therapeutic response to Adriamycin and 5-Fluorouracilboth in vitro and in vivo [9]. Corroborative analyses with a panel of CRC lines (Colo205, Colo320, SW-620, SW-480) revealed MKK3 as an essential surviving factor in most tested lines [1]. Furthermore, our studies demonstrated that MKK3 counteracts 5-FU efficacy by activating specifically the p38δ MAPK (MAPK11) isoform, inducing ERCC1 expression. Indeed, the MKK3 silencing abrogates the p38δ MAPK protein levels [1], and the p38δ MAPK depletion (siRNA) abrogates ERCC1 mRNA and protein levels, enhancing 5FU cytotoxicity significantly in all tested CRC lines. Overall, these findings suggest novel potential therapeutic approaches in CRC [35]. Importantly, similarly to primary epithelial breast cells MCF10A [9], siRNA abrogation of MKK3 was well-tolerated in primary colonocytes (CCD-841 and CCD-18-CO) [1], further supporting MKK3 as a potential therapeutic antitumor target [36].

## 3. Oncosuppressive MKK3 Functions

### 3.1. Glioblastoma

Zhu et al. [37,38] reported the activation of the signaling pathway GMFβ-MKK3/6-p38MAPK in glioblastoma to be involved in the β-elemene-induced G0/G1 cell cycle arrest in human U87 and rat C6 glioblastoma cells, as either the GMFβ (Glia maturation factor β) silencing (siRNA) or MKK3 and MKK6 inhibition (dominant negative vectors), which hampered the β-elemene-induced effects [37,38]. The β-elemene sensitizes glioblastoma cells to cisplatin through GMF-β activation [38], as the high GMF-β and p38MAPK activation enhances the killing efficacy of cisplatin in glioblastoma cells [38]. GMF-β depletion affects the β-elemene antiproliferative effects in vitro hampering MKK3 activation [38], uncovering a novel molecular mechanism involving MKK3 and GMF-β in the β-elemene driven inhibition of glioblastoma cell proliferation.

### 3.2. Breast Cancer

In breast cancer (Figure 1A), MacNeil et al., described the reduced MKK3 gene copy number associated with tumor malignancy. Indeed, the MKK3 overexpression inhibits breast cancer cell proliferation by inducing G1-phase cell cycle arrest through the transcriptional upregulation of cyclin-dependent kinase inhibitors (p21 and p27) in MDA-MB-468 cells [39]. Tsai et al., reported MKK3 tumor suppressor functions in estrogen receptor-positive (ER+) breast cancer MCF7 and BT474 lines [40], demonstrating that activation of the Rac1-MKK3/6-p38MAPK signaling is required for Connexin 43 (Cx43) protein reduction induced by Fulvestran and Tamoxifen treatments in ER+ cells inhibiting migration and invasion. Inhibition of either Rac1 (EHOP-016), p38MAPK (SB203580), MKK3, or MKK6 (siRNA) prevents the Nedd4-mediated ubiquitination and proteasome degradation of Cx43 promoting cell migration [40].

### 3.3. Lung Cancer

In lung cancer studies (Figure 2B), Samulin Erdem et al. [41] screened tumoral and normal lung tissue from 233 NSCLC patients for *MKK3* and MAPK-activated kinase 2 (*MK2*) genes copy number alterations (CNAs). They detected copy number loss in MKK3 (31%) and MK2 (28%) in NSCLC when compared with only 7% in healthy adjacent tissues [41]. Furthermore, the loss of MKK3 copy number was significantly more frequent in squamous cell carcinoma (SQ) and large cell carcinoma (LC) than in adenocarcinoma (AD), suggesting the potential value of MKK3 as a novel tumor suppressor in NSCLC [41].

### 3.4. Gastric Cancer

Zhang et al. [42] found dehydroeffusol (DHE) in gastric cancer to induce tumor-suppressing endoplasmic reticulum (ER) stress activating the DNA damage-inducible transcript 3 (DDIT3) transcription factor via the MEKK4-MKK3-p38MAPK signaling pathway activation inhibiting MGC803 cell proliferation. DHE activates the MKK3-p38MAPK signaling selectively, and SB203580 significantly inhibited the DHE-induced DDIT3 upregulation and apoptosis in MGC803 [42].

### 3.5. Liver Cancer

In hepatocellular carcinoma (Figure 2A), Wang et al. [43] described MKK3 oncosuppressive functions since its overexpression impaired proliferation and G1 phase cycle arrest by upregulating the CDK4/6 inhibitors (p16INK4A and p15INK4B) in HepG2 and PLC-PRF-5 cells. The MKK3 tumor-suppressing role depended on Bmi-1 downregulation and p38MAPK activation, as the SB203580 rescued Bmi-1 expression downregulating p16 INK4A and p15 INK4B to normal levels and suppressing the MKK3-induced cell cycle arrest in HepG2 and PLC-PRF-5 cells [43]. Lu et al. [44] analyzed the clinical pathological data of 75 HCC patients and correlated the overexpression of ATPase family AAA domain-containing protein 2 (ATAD2) with more aggressive phenotypes since significantly associated with high AFP levels, advanced tumor stages, and vascular invasion [44]. The ATAD2 knockdown (siRNA, shRNA) decreases cell viability, colony formation, migration, and invasion in HepG2, Hep3B, Huh7, and PLC/PRF/5 cells bearing high endogenous levels of ATAD2. The p38MAPK pathways activation is required to induce apoptosis in ATAD2 depleted cells. Mechanistically, ATAD2 directly interacts with MKK3/6 preventing p38MAPK activation and thereby inhibiting p38MAPK-induced apoptosis [44]. Xu et al. [45] by investigating the miR-21 molecular mechanisms involved in supporting hepatocellular carcinoma HEPG2 cell proliferation, reported by immunohistochemistry staining on 14 tumor samples, the miR-21 expression inversely correlated with MKK3 staining in tumor tissues, which expression was strikingly repressed when compared to relative adjacent non-tumor tissues [45]. The Study identified MKK3 as a novel direct target of miR-21, correlating its repression as novel regulatory feedback to induce HEPG2 cell proliferation. However, although of interest, no specific MKK3 approaches have been adopted in these studies [44,45], thus not allowing to assess the oncosuppressive MKK3 roles in these experimental models.

### 3.6. Cervical and Ovarian Cancer

In cervical cancer (Figure 1B), Lee et al. [46] analyzed the molecular mechanisms involved in alpha-mangostin anticancer activities and reported ROS-dependent ASK1/MKK3/6-p38MAPK signaling activation is necessary to induce apoptosis in HeLa and SiHa cells. The knockdown of either MKK3/6 or ASK1 significantly reduced the α-mangostin-induced apoptosis [46]. However, the lack of specific MKK3 inhibition does not allow us to assess its oncosuppressive roles in these experimental settings. By investigating the molecular mechanisms through which the metastasis-associated protein 2 (MTA2) regulates matrix metalloproteinase 12 (MMP12) in mediating cervical cancer metastasis, Lin et al. [47] identified a negative regulatory crosstalk ASK1/MKK3-p38MAPK/p-Y-box binding protein 1 (YB1) that activated upon MTA2 silencing, and disrupted the binding of AP1 (c-Fos/c-Jun) to MMP12 promoter, thus hampering tumor invasion [47].

In ovarian cancer, Mansouri et al. [48] dissected the roles of MKK3-p38MAPK signaling in driving response to cisplatin, demonstrating that the p38MAPK inhibition (SB202190) strongly hampers cisplatin-induced cell death, whereas ectopic expression of constitutive activated MKK3 induces apoptosis [48].

### 3.7. Esophageal Cancer

Han et al. [49] investigated the molecular mechanisms behind the Sulforaphene (SFE) antiproliferative effects in esophageal cancer cells in vitro (EC109, KYSE510, KYSE150, TE-1) and xenograft model (KYSE150) in nude mice. They identified an SFE dependent regulatory positive feedback loop GADD45B-MKK3-p38MAPK-p53-GADD45B. The authors demonstrated that GADD45B, a direct target of SFE, combines with MAP3K4, thereby activating MKK3. The specific MKK3 siRNA depletion contributes to rescuing the cell apoptosis and G2/M arrest induced by SFE, whereas MKK3 ectopic expression inhibited cancer cell proliferation [49]. Zhang et al., demonstrated the miR-19b-3p/MKK3/STAT3 feedback loop involved in esophageal tumorigenesis [50]. The authors showed that ectopic MKK3 expression significantly decreased cell proliferation, colony-forming ability, and invasion, inducing apoptosis in KYSE150 and KYSE520 esophageal cancer cells, whereas MKK3 siRNA depletion or CRIPS/Cas9 knockout induced opposite effects in vitro and in vivo in xenograft models in nude mice [50]. Furthermore, MKK3 overexpression decreased STAT3 transcriptional activity. Mechanistically MKK3 interacts directly with STAT3 inducing protein degradation via MDM2. Conversely, STAT3 induces miR-19b-3p expression that represses the MKK3 mRNA through binding its 3′UTR [50].

## 4. Conclusions

MKK3 was first described in 1996, and almost 700 MKK3 studies have been published since then. These publications mostly focus on MKK3′s well-known function as the main activator of p38 MAPK signaling, orchestrating a plethora of cellular processes. These include proliferation, differentiation, apoptosis, invasion, and cellular migration, in response to different cellular stress, such as exposure to UV damage, DNA damaging agents, oxidative stress, and growth factors and cytokines. Emerging research points to novel MKK3 functions based on its ability to interact with several partners identifying MKK3 as a hub protein [7]. However, since MKK3 is not frequently mutated in human cancer, the consequences of its dysregulation in gene expression or protein activity in cancer are still not clearly defined.

Ongoing studies suggest MKK3 plays an oncogenic role; identified among the upregulated target genes by p53 gain-of-function mutants, its specific knockdown significantly affects proliferation and survival in cancer lines of different tumor types (breast, colon, melanoma), both in vitro and in vivo by inducing autophagy and cell death. Interestingly, our studies revealed that MKK3 is superfluous in an untransformed cellular context (primary mammary epithelial cells, colonocytes), as its knockdown is well tolerated. Those data follow the in vivo studies that revealed MKK3-null mice viable and fertile [51], suggesting MKK3 as a potential novel target for tumor therapy. Contrarily, parallel literature reported MKK3 oncosuppressive function, which could not be ignored.

Accordingly, to provide a comprehensive picture of the MKK3 function in human cancers, we systematically analyzed the recently published literature (last almost 20 years), highlighting oncogenic and oncosuppressive reported MKK3 functions in the different tumor contexts. Interestingly, the analysis revealed that the oncogenic and tumor-suppressing roles of MKK3 are clearly dependent on the tumor type. Our analysis conclusively reported MKK3 oncogenic functions in melanoma, prostate, and colon cancer (Table 1), while oncosuppressive functions were identified in glioblastoma and gastric cancer (Table 1). Moreover, dual MKK3 oncogenic and oncosuppressive functions were found in breast, cervical, ovarian, liver, esophageal, and lung cancer (Table 1, Figure 1 and Figure 2), although relevant literature predominantly describes the oncogenic function in these diseases, especially in cervical, ovarian, and lung cancer.

The latest research in the MKK3 role is further assessing its oncogenic functions, including the study conducted by Sun et al., who assessed MKK3 licorne (lic) as an essential regulator of JNK-mediated cell migration and invasion. The authors used genetic screening based on the loss-of-cell polarity-triggered cell migration in the wing epithelia in Drosophila. Indeed, the Lic ectopic expression was sufficient to induce JNK-mediated, p38MAPK-independent cell migration, cooperating with oncogenic RAS to promote tumor invasion. This demonstrates that, besides its well-known role as a p38MAPK activator, MKK3 is involved in novel regulatory networks controlling cellular processes such as the JNK-dependent cell migration and invasion [52]. In accordance, Behren et al., identified FOXM1 as a key downstream target of RAS- and MKK3- induced cellular in vitro invasion and anchorage-independent growth signaling in NIH3T3 cells [53]. Immunohistochemical staining on tumor samples and adjacent normal tissues is providing new insights into the MKK3 roles in these diseases. Although the literature indicates oncosuppressive MKK3 functions [37,38], the reported immunohistochemical staining on tumor and normal brain tissues revealed the expression of MKK3/6 and p-MKK3/6 significantly higher in glioblastoma than in the normal brain tissues [37], suggesting that more investigations are required to define the MKK3 roles in this pathology. Accordingly, Meng et al., analyzed clinical samples of high-grade serous ovarian cancer and adjacent non-tumor tissues. They showed that high expression of MAP2K3 was significantly correlated with CA125 level (*p* < 0.001), tumor size (*p* = 0.001), lymph node metastasis (*p* = 0.008), depth of myometrial invasion (*p* < 0.001), and FIGO stage (*p* < 0.001), indicating MKK3 as a poor prognostic biomarker [54]. Conceicao et al., conducted immunohistochemical staining on 24 samples of giant cell tumors of bone and normal tissues and reported higher expression of the MKK3, MMP14, TIMP2, and VIM genes in tumor than the normal counterpart, suggesting their involvement in invasion and metastasis [55]. With the tissue microarray (TMA) performed in a cohort of 189 colorectal cancer patients, our study identified MKK3 as a poor prognostic marker, correlating significantly high MKK3 staining with short overall survival (OS) in late-stage CRC patients [1].

Our ongoing studies support the abrogation of MKK3 functions as a novel therapeutic strategy against cancer. Through an inducible RNAi lentiviral-based experimental system, we were able to assess, in breast and colon cancer lines, the MKK3 contribution in sustaining cancer cell proliferation, survival, invasion, as well as hampering the response to chemotherapeutic treatments [1,9] and tumor malignancy with xenograft models in nude mice [1,8,9]. Moreover, in adopted experimental models, the MKK3 depletion enhances response to chemotherapeutic treatments allowing the drug-dose reduction, thereby preserving the effectiveness of the therapy. Overall, following results achieved in this review, MKK3 targeting could represent a novel promising molecular target for developing new anticancer therapies.

No specific MKK3 inhibitors are currently available; thus, identifying the strategies for targeting MKK3 is challenging. Recent studies identified MKK3 as hub protein due to its interaction with several molecular partners, including MYC [7]; this interaction was reported to sustain MYC stability supporting breast cancer cells proliferation [7]. Accordingly, disruption of MKK3-MYC protein–protein interaction has been suggested as a new strategy to target MYC-driven action [56]. Ongoing studies involving the drug repositioning approach have identified FDA-approved compounds that enable us to recapitulate MKK3 depletion-dependent effects in different CRC lines both in vitro and xenograft models in nude mice, thus identifying novel promising drugs to potentially translate from bench to bedside. Insights gained thus far suggest a more complex picture of MKK3′s involvement in tumor malignancy. Thus, exhaustive investigations assessing the MKK3 functions in different tumor types are imperative, especially studies targeting its novel identified ability to interact, as a hub protein, with several molecular partners. Finally, this systematic analysis suggests that MKK3 must be contextualized within a given tumor type in order to adequately evaluate its potential efficacy as a molecular target for developing novel anticancer therapies.

**Table 1 cancers-14-00483-t001:** Pathologies with reported MKK3 oncogenic and tumor-suppressing functions and regulatory partner involved.

Disease	Oncogenic Functions Regulatory Partners	References	Tumor-Suppressing Functions Regulatory Partners	References
breast cancer	MEK2/Cyclin D1 *; RAS/MMP2 *; RORO2/MMP2 *; ROR2/MMP9 *; ROR2/TGF-β, TCTP/SCR *, c-MYC	[2,3,7,8,9,10,11,12,13,14]	p21^Cip1^, p27^Kip1^; RAC1/Cx43 *	[39,40]
cervical and ovarian cancer	miR214; OPN/CD44/NF-KB/FURIN *; MDR1; MMP2; MMP9; CDK4; CDK2; Integrin β1	[15,16,17,18,19]	ASK1; ASK1/*p*-YB1*	[46,47,48]
liver cancer	SIRT1/YAP *; SPAG9	[20,21]	Bmi1-p16INK4A, -p15INK4B; ATAD2, miR-21	[43,44,45]
melanoma	miR-21; L1CAM	[22,23]		
prostate carcinoma	GRP78/α2M *; TRAIL/MCL1 *; SRC/SF *	[24,25,26]		
esophageal cancer	MKK3/6-p38MAPK	[27]	GADD45/p53; miR19b-3p/STAT3	[49,50]
lung cancer	p38MAPK; ERCC1; MSH2; STAT3	[28,29,30,31,32,33,34]	MKK3 CNA	[41]
colorectal cancer	MAPK11-ERCC1	[1,8,9,35]		
glioblastoma			GMF-β	[37,38]
gastric cancer			DDIT3	[42]

* bars indicating upstream and downstream effectors of MKK3.

## Figures and Tables

**Figure 1 cancers-14-00483-f001:**
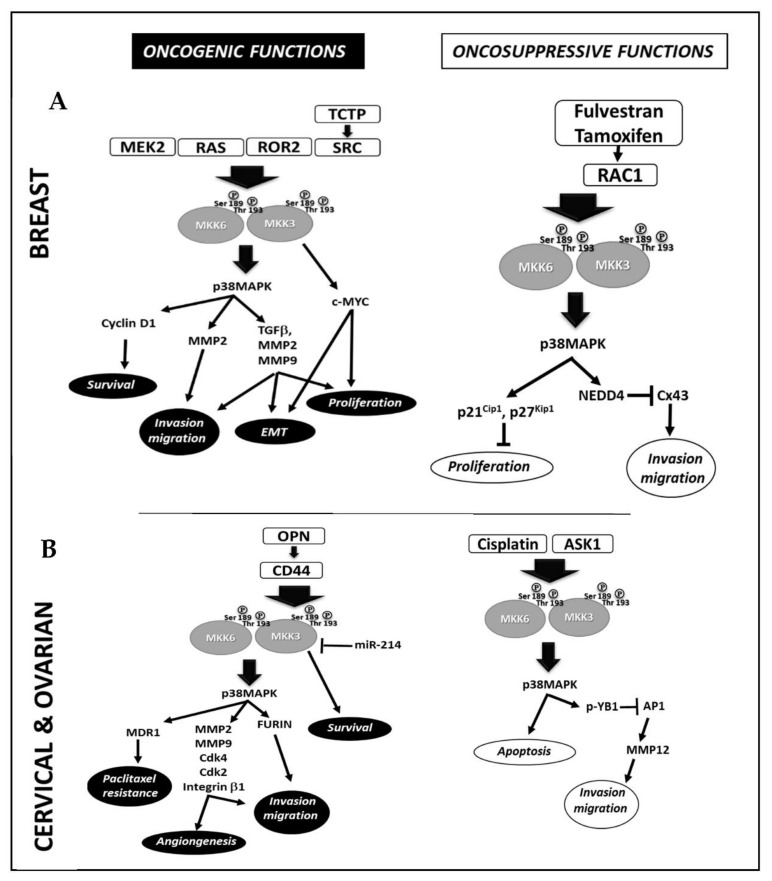
Molecular mechanisms involved in MKK3 oncogenic and tumor-suppressive functions in Breast (**A**) and Cervical and Ovarian cancer (**B**).

**Figure 2 cancers-14-00483-f002:**
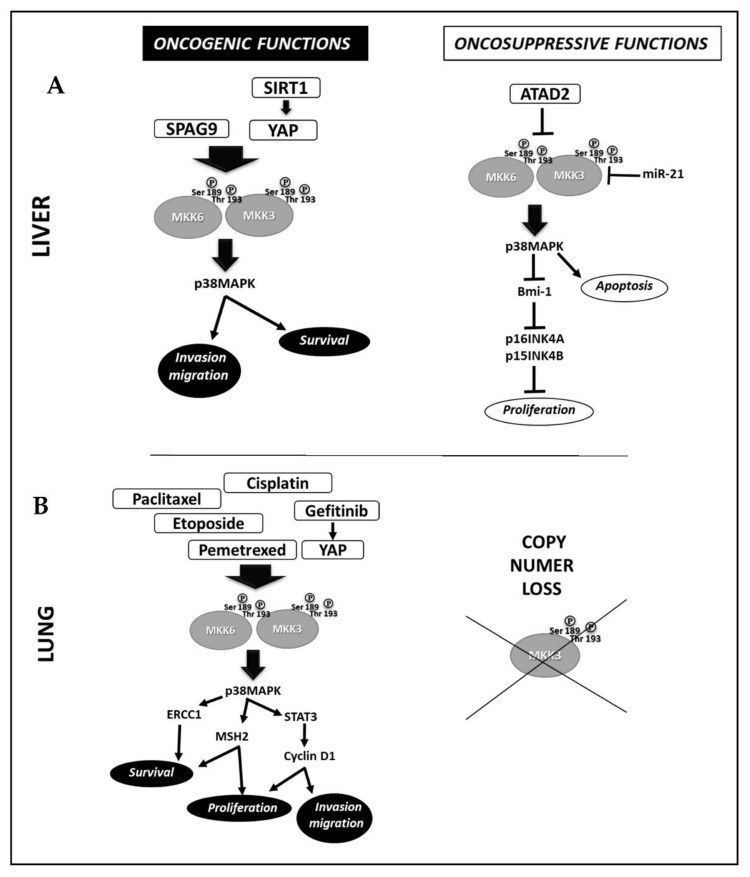
Molecular mechanisms involved in MKK3 oncogenic and tumor-suppressive functions in Liver cancer (**A**) and Lung cancer (**B**).

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
