# Peer review of "Dissection of the MKK3 Functions in Human Cancer: A Double-Edged Sword?"

_cancers, 2022, doi:10.3390/cancers14030483_

Round 1

Reviewer 1 Report

This review aims to summarize recent and relevant publications about MKK3 an evolutionarily conserved protein kinase, that is activated in response to a variety of stimuli and phosphorylates specifically the p38MAPK family proteins contributing in the regulation of a plethora of cellular processes. While such an effort may be a worthwhile when done in a systematic scientific approach the endeavour presented in this review is not.

It is not even documented which search term where used, how many publications where found (Over 700?), how the data where extracted etc. etc.

So it adds little value or help people working in the field, while for the general audience it is of little interest since it is only a listing of the involvement of MKK3 in various tumor types and its so far unclear functional relevance for onogenesis.

Author Response

It is not even documented which search term where used, how many publications where found (Over 700?), how the data where extracted etc. etc.

Thanks for suggestion, pubmed searching was performed for “MKK3 & cancer” (https://pubmed.ncbi.nlm.nih.gov/?term=%22mkk3+%26+cancer%22&sort=pubdate) or “MAP2K3 & canbcer” (https://pubmed.ncbi.nlm.nih.gov/?term=%22MAP2K3+%26+cancer%22) identified respectively 158 and 102 manuscripts from 1996 to 2021. Fifty-five out of all analyzed manuscripts reported specific MKK3 functions were selected.  This paragraph has been added to integrate the introduction paragraph.

So it adds little value or help people working in the field, while for the general audience it is of little interest since it is only a listing of the involvement of MKK3 in various tumor types and its so far unclear functional relevance for onogenesis.

This work aims to define the state of the art on the knowlegment of MKK3 roles in cancer, since literature reported both tumor suppressive and oncogenic functions its role in not well understood. Outcomes of this analysis is pointing out for the first time the organ specific contribution of MKK3 in tumor progression or repression, underlining the more complex role of MKK3 in cancer besides its wel-known role as p38MAPK activator.  

Reviewer 2 Report

In this submission, Piastra highlighted that the oncogenic or tumor suppressive role of MKK3 is strictly dependent on the tumour type, suggesting a more complex scenario involving MKK3 in tumour malignancy. This is an interesting study, but a few mechanisms are not clear in its current format. I recommend this paper to be accepted with subject to major revisions. 

  • abstract - most of the content is dedicated to background, only one last sentence reveal what this review highlights. Please add an appropriate discussion around this review and why is this review important and how this review addresses the solutions, promises, limitations and challenges. and at the end how this is applicable in terms of clinically relevant endpoints.
  • Introduction - a comparison to existing reviews should be given very clearly.
  • An independent section on discussion should be included where authors should include a detailed table on the whole technology with key parameters, cancer type, in vitro, in vivo models, etc.
  • In vivo model description is limited, authors should give some inisghts into this too.

Author Response

abstract - most of the content is dedicated to background, only one last sentence reveal what this review highlights. Please add an appropriate discussion around this review and why is this review important and how this review addresses the solutions, promises, limitations and challenges. and at the end how this is applicable in terms of clinically relevant endpoints.

Thanks for suggestion, abstract has been modified accordingly in the revised manuscript version

Introduction - a comparison to existing reviews should be given very clearly.

Thanks for suggestion, introduction has been integrated accordingly in the revised manuscript version

An independent section on discussion should be included where authors should include a detailed table on the whole technology with key parameters, cancer type, in vitro, in vivo models, etc.

Thanks for suggestion, the revised manuscript has been integrated with two novel figures reporting the network MKK3-dependent involved for those diseases that reported a dual function of MKK3, moreover the Table 1 has been integrated with two more columns describing the key players reported to support either MKK3 oncosuppressive of oncogenic functions. We hope these integrations could partially address what asked by this reviewer.

In vivo model description is limited, authors should give some inisghts into this too.

The in vivo studies described are mainly with xenograft model in nude mice, in the revised manuscript description of in vivo models are reported along the text.

Reviewer 3 Report

The enclosed manuscript entitled "Dissection of the MKK3 Functions in Human Cancer: A Double-Edged Sword" submitted by Piastra et al. intends to introduce the role of MKK3 in various cancers. MKK3 is widely recognized as an oncogene in many cancers; however, elevating evidence suggests MKK3 may play a role as a tumor suppressor in some other types of cancers as well. Given that MKK3 in GBM and gastric cancer are considered tumor-suppressive, MKK3 shows bi-phasic behavior between oncogenic and tumor-suppressive in some cancers, such as lung cancers and ovarian cancers. In general, this draft is written at a reasonable pace and structure with sufficient inclusive reference and elaboration; however, some issues may achieve a better scientific impact eventually. 

1) MKK3 is serving as an upstream activator to MAPK p38, and it is commonly involved in many cancers in the oncogenic route. Other events may counteract this mechanism such as miRNA, transcription factors, and some stress inducers. In this regard, it might be better to use an illustrative scheme to map the interaction between molecules. Especially, the MKK3-MAPK seems to have both oncogenic and tumor-suppressive features in some cancers but it is confusing with only a text description. 

2) It is good to have a graphical expression for the dominance of MKK3 in various cancers; however, it is fairly strange to justify the role of MKK3 in each cancer by the number of references found. If so, the authors may need to provide a comprehensive literature searching strategy and inclusive/exclusive criteria. 

3) Apart from the review of the respective regulatory routes of MKK3 in different cancers, I didn't find any clues in the "double-edged sword" if we use MKK3 as a target of anti-cancer treatment. 

4) the table is not in a well-arranged manner, and it is expected to involve some columns for the role of MKK3 and the regulatory partners to make the table more informative. 

Author Response

1) MKK3 is serving as an upstream activator to MAPK p38, and it is commonly involved in many cancers in the oncogenic route. Other events may counteract this mechanism such as miRNA, transcription factors, and some stress inducers. In this regard, it might be better to use an illustrative scheme to map the interaction between molecules. Especially, the MKK3-MAPK seems to have both oncogenic and tumor-suppressive features in some cancers but it is confusing with only a text description.

Thanks for suggestion, accordingly illustrative schemes have been enclosed to integrate the revised manuscript for those disease where dual MKK3 roles where reported, with the novel Figures 1A,1B and 2A,2B.

2) It is good to have a graphical expression for the dominance of MKK3 in various cancers; however, it is fairly strange to justify the role of MKK3 in each cancer by the number of references found. If so, the authors may need to provide a comprehensive literature searching strategy and inclusive/exclusive criteria.

Thanks for this comment, with figure 1 we only aimed to summarize by illustrative scheme all the relevant data collected, we agree with this reviewer and accordingly figure 1 has been removed in the revised manuscript.      

3) Apart from the review of the respective regulatory routes of MKK3 in different cancers, I didn't find any clues in the "double-edged sword" if we use MKK3 as a target of anti-cancer treatment.

We thank the reviewer for provided comment, since the evidenced oncosuppressive and oncogenic functions in different tumor type, taken into account of MKK3 as target for anticancer treatments should be contextualized with tumor type, since approach aimed to inhibit MKK3 functions, as our studies are supporting, should be considered only for those tumors where oncogenic functions are reported not for those with oncosuppressive functions, this was the bottom line idea to consider MKK3 as “double-edged sword”

4) the table is not in a well-arranged manner, and it is expected to involve some columns for the role of MKK3 and the regulatory partners to make the table more informative.

Thanks for suggestion, accordingly table 1 has been integrated in the revised manuscript.

Reviewer 4 Report

The manuscript Dissection of the MKK3 Functions in Human Cancer: A Dou-ble-Edged Sword? by Piastra V et al. represents an interesting and well-structured review article that describe the oncogenic and oncosuppressive functions of MKK3 in several types of cancer.

There are few comments. In order to make the manuscript more understandable by the readers, I suggest to add along with the table 1 an additional table to describe the potential new therapeutic approaches based on targeting of MKK3 protein or its downstream signalling mediators and effectors involved in oncogenic pathways. Furthermore, it would be interesting add in the “Conclusion paragraph” some comments on the translational relevance that the new evidences on the role of MKK3 to mediate drug resistance, metastasis and invasion may have on the treatment of cancer patients. In this respect, I suggest to add more recent references describing in vivo studies in the such research field and report in the text some images (if possible).

Author Response

There are few comments. In order to make the manuscript more understandable by the readers, I suggest to add along with the table 1 an additional table to describe the potential new therapeutic approaches based on targeting of MKK3 protein or its downstream signalling mediators and effectors involved in oncogenic pathways.

Thanks for suggestion, the revised manuscript has been integrated with two novel figures ( Figure1A,B; 2A,B) reporting the networks MKK3-dependent involved for those diseases that revealed dual functions of MKK3. Moreover, the Table 1 has been integrated with two more columns describing the key players upstream and downstream to MKK3 reported to support either oncosuppressive of oncogenic functions. We hope these integrations could partially address what asked by this reviewer.

Furthermore, it would be interesting add in the “Conclusion paragraph” some comments on the translational relevance that the new evidences on the role of MKK3 to mediate drug resistance, metastasis and invasion may have on the treatment of cancer patients. In this respect, I suggest to add more recent references describing in vivo studies in the such research field and report in the text some images (if possible).

Thanks for suggestion, in the revised manuscript the “conclusion paragraph” has been integrated with comments related the translational relevance of MKK3 as molecular target for design of novel anticancer therapies.

Round 2

Reviewer 2 Report

I am pleased to recommend the revised manuscript for publication in Cancers.

Reviewer 3 Report

The authors have addressed all my comments comprehensively.